# communications
## engineering

# A generalized dual-domain generative framework with hierarchical consistency for medical image reconstruction and synthesis

Jiadong Zhang [1,6], Kaicong Sun[1,6], Junwei Yang[1,2,6], Yan Hu[1,3,6], Yuning Gu[1,6], Zhiming Cui[1,6], Xiaopeng Zong[1], Fei Gao [1] & Dinggang Shen[1,4,5 ✉]

Medical image reconstruction and synthesis are critical for imaging quality, disease diagnosis and treatment. Most of the existing generative models ignore the fact that medical imaging usually occurs in the acquisition domain, which is different from, but associated with, the image domain. Such methods exploit either single-domain or dual-domain information and suffer from inefficient information coupling across domains. Moreover, these models are usually designed specifically and not general enough for different tasks. Here we present a generalized dual-domain generative framework to facilitate the connections within and across domains by elaborately-designed hierarchical consistency constraints. A multi-stage learning strategy is proposed to construct hierarchical constraints effectively and stably. We conducted experiments for representative generative tasks including low-dose PET/CT reconstruction, CT metal artifact reduction, fast MRI reconstruction, and PET/CT synthesis. All these tasks share the same framework and achieve better performance, which validates the effectiveness of our framework. This technology is expected to be applied in clinical imaging to increase diagnosis efficiency and accuracy.

[1] School of Biomedical Engineering, State Key Laboratory of Advanced Medical Materials and Devices, ShanghaiTech University, 201210 Shanghai, China. [2] Department of Computer Science and Technology, University of Cambridge, Cambridge CB2 1TN, UK. [3] School of Computer Science and Engineering, The University of New South Wales, Sydney, NSW 2052, Australia. [4] Shanghai United Imaging Intelligence Co., Ltd., 200230 Shanghai, China. [5] Shanghai Clinical Research and Trial Center, 200052 Shanghai, China. [6]These authors contributed equally: Jiadong Zhang, Kaicong Sun, Junwei Yang, Yan Hu, Yuning Gu, Zhiming Cui. ✉email: Dinggang.Shen@gmail.com

Medical imaging as an indispensable imaging technique plays a critical role in plenty of clinical applications, such as screening, disease diagnosis, and treatment planning. The enhancement of image quality has been a central topic for decades in the field of medical image processing[1]. Different from natural images which are usually captured directly in the image domain, medical imaging usually acquires data in the modality-specific domain such as the $k$-space domain for magnetic resonance imaging (MRI)[2], and the sinogram domain for computed tomography (CT)[3] and positron emission tomography (PET)[4]. To take advantage of the acquisition property of medical imaging, one should exploit the underlying information patterns in both the acquisition domain and the image domain. Moreover, one can apply more sophisticated constraints such as cycle consistency within and across domains to better regularize the solution space for generative tasks.

Medical image reconstruction and synthesis are the typical generative tasks in medical imaging, and can strongly benefit from the aforementioned dual-domain cycle-consistency scheme. Medical image reconstruction is one of the pillars of medical imaging. Reconstruction tasks usually can be categorized into two subgroups[5]: (1) reconstruction in the form of forward/backward transform such as low-dose CT[6–8] and fast MRI reconstruction[9–11]; (2) reconstruction as post-processing to improve image quality such as metal artifact reduction (MAR)[12–14] and super-resolution (SR)[15–17]. Most of the recent dual-domain-based medical image reconstruction works exploit dual-domain information by individual sub-networks, which are connected either in parallel branches[18,19] or sequentially in a cascaded manner[20–22], which can be further used for the diagnosis tasks[23,24]. The backbone of the sub-networks can be UNet-like[19,21,22,25], Transformer[26], or recently emerged Diffusion model[27]. In particular, Jun et al.[19] utilize two parallel UNet-shaped networks serving as regularizations in the $k$-space and image domains, respectively, for MRI reconstruction. Reseachers[21] also adopt sequentially cascaded sinogram and image networks for simultaneous metal artifact reduction and low-dose CT reconstruction. Although these existing reconstruction methods[20,28–31] have taken dual-domain information into account for better data consistency and overall performance, to our best knowledge, these networks are task-specifically designed and there is limited work that uses a generalized framework for dual-domain reconstruction. More importantly, there is no study yet considering structured consistency constraints which cover multi-level constraints within and across domains for better regularization of the solution domain.

Different from image reconstruction, medical image synthesis aims to infer a desired imaging modality without an actual scan such as synthesizing imaging modalities which are usually unavailable in routine clinical practice, i.e., due to cost. Further, since different modalities can reveal complementary physical characteristics of the underlying tissues, synthesis of missing or key modalities can lead to more accurate diagnosis and treatment planning[32,33]. Medical image synthesis can be categorized into inter-modality synthesis and intra-modality synthesis[32]. The inter-modality synthesis denotes image synthesis between two imaging modalities such as from PET to CT[34], while the intra-modality counterpart refers to the studies such as transferring between different MRI sequences[19]. Most of the existing learning-based models for medical image synthesis adopt VAE[35], GAN[36]-based network architecture, and its variations[37–39]. By resorting to the adversarial learning strategy, they obtain more plausible and real-looking images. For example, Dong et al.[40] propose a framework based on cycle-consistent generative adversarial networks (CycleGAN) to synthesize CT images from the non-attenuation corrected PET. Another method MedGAN[38], as a non-application-specific framework, merges the adversarial framework with several feature-level similarity metrics to facilitate the similarity match. However, these representative methods, following the works for natural images, purely manipulate information in the image domain, without considering inherent differences in image acquisition between natural images and medical images. And, ignoring dual-domain cycle consistency limits their performance in medical image synthesis.

To cope with the above-mentioned issues, we present a generalized learning-based dual-domain framework for generative tasks of medical images, by employing hierarchical consistency constraints including all possible directional constraints within and across domains by means of a multi-stage learning strategy. To be specific, different from the majority of existing generative models that manipulate in a single domain such as CycleGAN[40], we propose to exploit the underlying patterns in both domains. Moreover, we aim to build up multi-level similarity constraints between the source and target images in dual domains to better regularize the solution space. As shown in Fig. 1a and b, without loss of generality, for reconstruction or synthesis tasks of medical imaging, there exist two domains, namely the image domain and the acquisition domain. The source and target images can be transformed bidirectionally in each domain based on their individual generative function $G$. Depending on the applications, function $G$ can be realized by learning-based or model-based functions. Cross-domain information can be exchanged by means of the transform function $F$ and the inverse transform function $F^{-1}$. The methods can be applied to various tasks, as shown in Fig. 1c. Inspired by CycleGAN[40], we introduce a multi-stage learning strategy S1–S3 which, respectively, accounts for intra-domain consistency, inter-domain consistency, and cycle consistency as demonstrated in Fig. 1d. These hierarchical intra- and inter-domain consistency constraints construct and consolidate a comprehensive multi-level similarity match between the source and target images. Particularly, the intra-domain consistency stage S1 imposes the primary consistency constraint within the individual domain. To preserve the inter-domain consistency, in stage S2 we introduce a sequential constraint as marked by black arrows. Lastly, to further strengthen the similarity match, in stage S3 we adopt the cycle-consistency constraint within and across domains as depicted by yellow and green arrows. For a detailed description, please refer to the "Methods" section. We have conducted quantitative and qualitative analyses of the proposed framework from different aspects. Our framework achieves remarkable performance gain in many representative generative tasks in medical imaging including low-dose PET and CT reconstruction, MAR, fast MRI reconstruction, and PET-CT synthesis.

## Results

To validate the effectiveness and generalizability of our generative framework, we carried out extensive experiments for representative reconstruction and synthesis tasks for medical imaging including low-dose PET/CT reconstruction, CT metal artifact reduction, accelerated MRI reconstruction, and PET-CT synthesis quantitatively and qualitatively. In Figs. 2 and 3, we demonstrate the performance of our framework for the above generative tasks in different cases. In Figs. 4 and 5, we perform an in-depth analysis of our framework using the noise power spectrum (NPS) calculated according to the work of Dobbins III et al.[41]. An elaborate investigation of the proposed framework is conducted, including quantitative and qualitative evaluation, effectiveness, and ablation studies. In the following experiments, we employ structural similarity index measurement (SSIM), peak signal-to-noise ratio (PSNR), NPS, and standardized uptake values (SUV) for quantitative assessment.

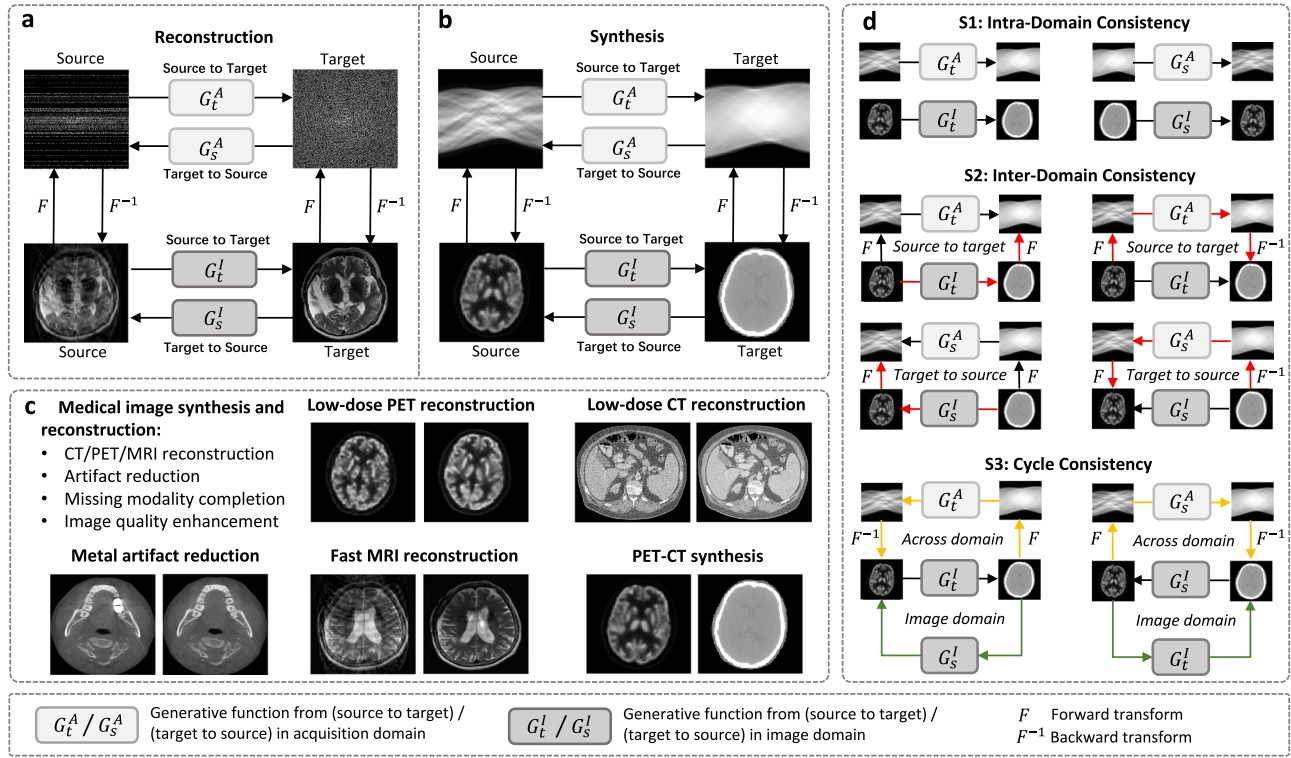

**Fig. 1 Overview of our proposed dual-domain generative framework with hierarchical consistency for medical image reconstruction and synthesis.**
**a** Framework for fast magnetic resonance imaging (MRI) medical reconstruction task. **b** Framework for positron emission tomography (PET) and computed tomography (CT) synthesis task. **c** Applications of the proposed generative framework, including low-dose PET/CT reconstruction, metal artifact reduction, fast MRI reconstruction, and PET-CT synthesis. **d** The proposed multi-stage training strategy, including Stage 1 (S1) intra-domain consistency, Stage 2 (S2) inter-domain consistency, and Stage 3 (S3) cycle consistency.

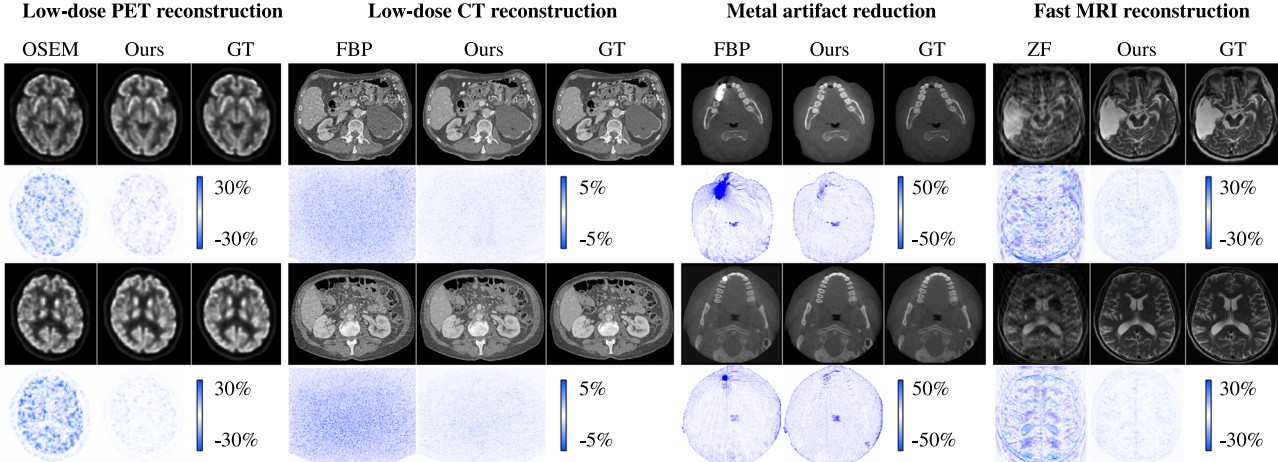

**Fig. 2 Application of the proposed framework on different reconstruction tasks.** For each of the tasks, we demonstrate the reconstructed images for two typical cases with their corresponding error maps compared to the ground-truth (GT) images. Ordered subset expectation-maximization (OSEM), filtered back projection (FBP), and zero filling (ZF) are traditional methods for PET, CT, MAR, and MRI reconstruction.

**Low-dose PET/CT reconstruction.** CT and PET both are imaging techniques based on ionizing radiation. To reduce radiation exposure risk in clinical applications, low-dose CT and low-dose PET are often preferred. Comparing to standard-dose CT and PET, the reconstruction of low-dose ones usually suffers from severe artifacts and noise, which may greatly affect a physician's diagnosis[42,43]. Hence, the development of effective reconstruction algorithms for low-dose PET/CT is a critical task that has an urgent demand in clinical practice. In this experiment, we evaluate the effectiveness of the proposed framework on low-dose

PET/CT reconstruction, where the Radon transform is used as the transform function $F$.

*Performance evaluation.* To evaluate the performance of our framework for low-dose PET/CT reconstruction, we compare the proposed reconstruction model with several representative learning-based reconstruction methods, including UNet[44], RUNet[45], p2pGAN[46], CycleGAN[40], and MedGAN[38], and also a recent dual-domain reconstruction method, i.e., iBP-Net[47]. We use the acquired low-dose images as source data and the

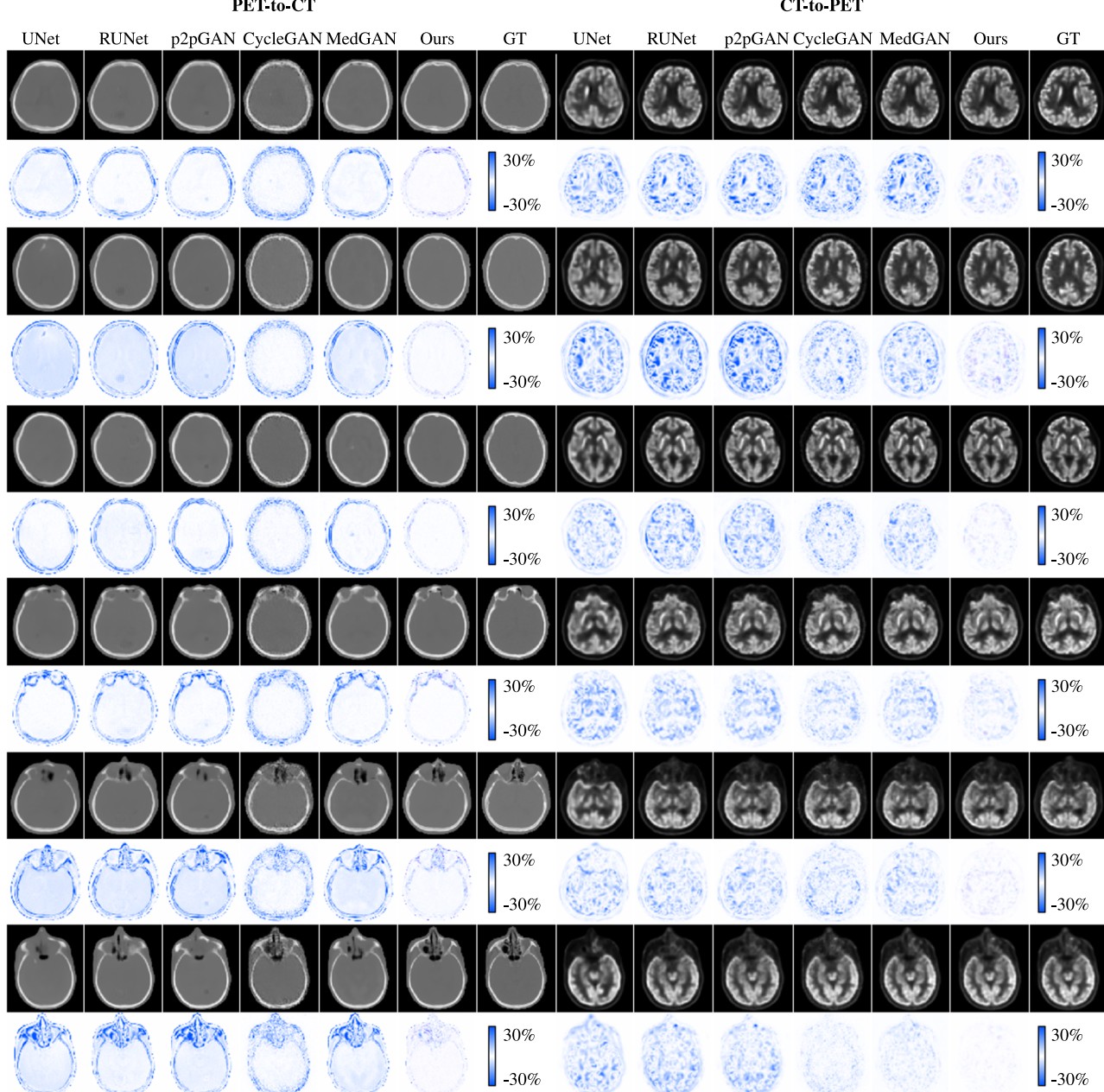

**Fig. 3 Visual comparison of synthesized positron emission tomography (PET) and computed tomography (CT) images for six typical cases by different methods.** The error maps are located beneath the corresponding reconstructed images. CT and PET images of the same row are paired.

standard-dose images as target data. The experimental results are summarized in Table 1. We can see that p2pGAN and MedGAN achieve better performance than RUNet by means of adversarial learning. CycleGAN slightly outperforms p2pGAN and MedGAN by using cycle-consistency. However, these methods only use image-domain information without considering sinogram patterns. Compared to the case of using only image-domain methods, iBP-Net combines the advantages of both domains, leading to better reconstruction performance than other comparison methods. On the other hand, iBP-Net reconstructs low-dose images using the cascaded dual-domain networks, which cannot explicitly guarantee dual-domain consistency. In contrast, our method exploits dual domains with multi-level consistency constraints and achieves the best performance in terms of both SSIM and PSNR metrics. To conduct a clinical evaluation of the reconstructed PET uptakes, we calculate SUV bias ($SUV_{mean}$ and

$SUV_{max}$) using the ground-truth standard-dose PET uptakes and report results in Table 1. These results show better performance of our method than other comparison methods.

*Effectiveness of dual-domain hierarchical consistency in low-dose PET/CT reconstruction.* The proposed generative framework contains multi-stage consistency constraints, consisting of intra-domain consistency (S1), inter-domain consistency (S2), and cycle-consistency (S3) to build up similarity match in a hierarchical manner between the source and target images within and across domains in bi-directions, i.e., from target to source and source to target. In this experiment, we perform an in-depth analysis of the effectiveness of the introduced dual-domain and hierarchical consistency scheme for low-dose PET/CT reconstruction. In particular, we evaluate the impact of each stage from S1 to S3 by ablation study on both reconstruction tasks. Besides,

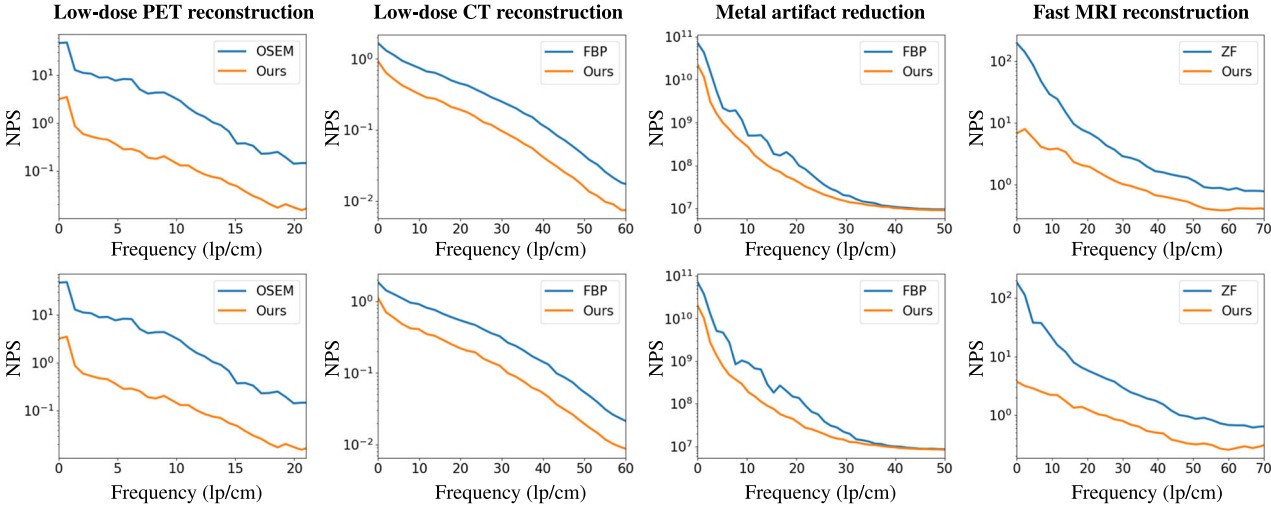

**Fig. 4 The noise power spectrum analysis (NPS) for different reconstruction tasks.** The error map between the reconstructed image and the ground-truth (GT) image is regarded as the noise image for NPS calculation.

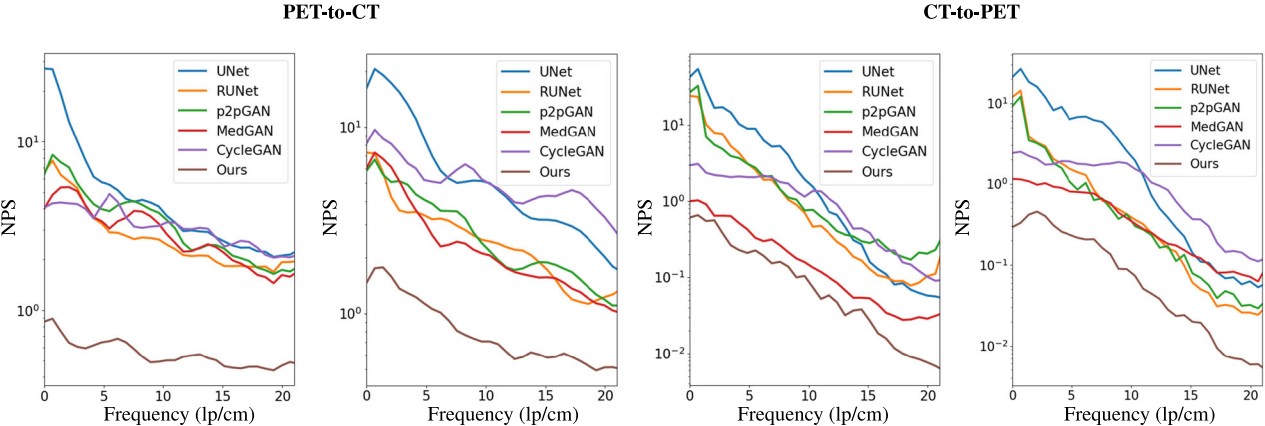

**Fig. 5 The noise power spectrum (NPS) analysis of the synthesized images by different state-of-the-art methods.** The error map between the synthesized image and the ground-truth (GT) image is used as the noise map for NPS calculation.

**Table 1 Quantitative comparison with representative learning-based methods for low-dose PET/CT reconstruction.**

| Tasks | Low-dose PET reconstruction | | | | Low-dose CT reconstruction | |
|---|---|---|---|---|---|---|
| Method | SSIM | PSNR | $SUV_{mean}$ | $SUV_{max}$ | SSIM | PSNR |
| RUNet[45] | 0.8064 ± 0.1028 | 35.07 ± 1.03 | 8.70 ± 6.79 | 6.23 ± 5.94 | 0.9311 ± 0.0823 | 41.02 ± 2.31 |
| p2pGAN[46] | 0.9844 ± 0.0129 | 37.55 ± 1.20 | 5.99 ± 9.04 | 3.92 ± 5.02 | 0.9712 ± 0.0510 | 42.41 ± 1.79 |
| CycleGAN[40] | 0.9861 ± 0.0078 | 37.85 ± 1.18 | 7.79 ± 5.86 | -6.63 ± 1.04 | 0.9811 ± 0.0086 | 41.90 ± 0.94 |
| MedGAN[38] | 0.9863 ± 0.0093 | 37.72 ± 1.11 | 4.01 ± 8.16 | -8.31 ± 4.59 | 0.9613 ± 0.0577 | 41.53 ± 2.05 |
| iBP-Net[47] | 0.9859 ± 0.0154 | 38.47 ± 1.28 | 6.84 ± 3.97 | 5.96 ± 1.79 | 0.9812 ± 0.0143 | 42.71 ± 0.97 |
| Ours | **0.9939 ± 0.0124** | **39.57 ± 1.03** | **3.62 ± 5.83** | **3.54 ± 2.40** | **0.9910 ± 0.0032** | **43.57 ± 0.85** |

Best performance is highlighted in bold font.

we conduct ablation study also for the cases of using single and dual domains.

Quantitative results are summarized in Table 2. Different consistency constraints are involved in the corresponding stages from S1 to S3 as shown in Fig. 1. From the table, we can see that there indeed exists noticeable performance improvement for both tasks when using hierarchical consistency constraints by multi-stage learning strategy. To evaluate the importance of the dual-domain scheme in the proposed generative framework, we carried out experiments based on PET/CT reconstruction tasks. From

Table 2, we can see that the dual-domain scheme achieves better performance than the single-domain as expected. The utilization of a dual-domain scheme allows the generative framework to better exploit the latent patterns in a complementary way. Moreover, it enables the integration of hierarchical consistency constraints into the framework.

**Metal artifact reduction.** Metal artifacts caused by the presence of metal implants such as dental fillings can generate abnormal

**Table 2 Ablation study of the proposed framework for low-dose PET/CT reconstruction including multi-stage and dual-domain schemes.**

| Setting | Low-dose PET reconstruction | | Low-dose CT reconstruction | |
|---|---|---|---|---|
| | SSIM | PSNR | SSIM | PSNR |
| S1 | 0.8064 ± 0.1028 | 35.07 ± 1.03 | 0.9311 ± 0.0823 | 41.02 ± 2.31 |
| S1 + S2 | 0.9788 ± 0.0274 | 37.22 ± 1.70 | 0.9771 ± 0.0138 | 42.02 ± 0.91 |
| S1 + S3 | 0.9816 ± 0.0225 | 37.51 ± 1.49 | 0.9762 ± 0.0167 | 42.49 ± 0.95 |
| S2 + S3 | 0.9837 ± 0.0201 | 38.25 ± 1.96 | 0.9835 ± 0.0137 | 42.95 ± 0.98 |
| S1 + S2 + S3 | **0.9939 ± 0.0124** | **39.57 ± 1.03** | **0.9910 ± 0.0032** | **43.57 ± 0.85** |
| Image domain | 0.9815 ± 0.0139 | 37.02 ± 1.41 | 0.9666 ± 0.0172 | 41.94 ± 1.81 |
| Dual-domain | **0.9939 ± 0.0124** | **39.57 ± 1.03** | **0.9910 ± 0.0032** | **43.57 ± 0.85** |

Best performance is highlighted in bold font.

**Table 3 Quantitative comparison with representative methods for MAR on both in-house teeth CBCT dataset and public DeepLesion dataset.**

| Method | CBCT | | DeepLesion | |
|---|---|---|---|---|
| | SSIM | PSNR | SSIM | PSNR |
| LI[48] | 0.8427 ± 0.0954 | 31.75 ± 2.89 | 0.9089 ± 0.0983 | 30.72 ± 2.75 |
| NMAR[49] | 0.9378 ± 0.0812 | 31.54 ± 2.68 | 0.9102 ± 0.0889 | 31.89 ± 2.01 |
| CycleGAN[39] | 0.8469 ± 0.2234 | 31.55 ± 2.97 | 0.7201 ± 0.1403 | 30.72 ± 1.97 |
| RCN | 0.9286 ± 0.1865 | 33.60 ± 1.69 | 0.9302 ± 0.1351 | 33.81 ± 1.44 |
| AttentionMAR[51] | 0.9507 ± 0.0537 | 36.25 ± 1.23 | 0.9421 ± 0.1056 | 34.71 ± 1.75 |
| DuDoNet[28] | 0.9572 ± 0.0492 | 36.74 ± 1.54 | 0.9492 ± 0.1123 | 35.23 ± 1.69 |
| Ours (RUNet-8) | 0.9328 ± 0.0573 | 35.02 ± 1.36 | 0.9203 ± 0.1518 | 33.76 ± 2.82 |
| Ours (RUNet-16) | 0.9599 ± 0.0487 | 36.84 ± 1.13 | 0.9327 ± 0.1275 | 34.79 ± 2.07 |
| Ours (RUNet-32) | **0.9634 ± 0.0295** | **36.91 ± 1.08** | **0.9502 ± 0.0872** | **35.93 ± 1.99** |

Best performance highlighted in bold font.

**Table 4 Ablation study of the proposed framework for MAR on both in-house teeth CBCT dataset and public DeepLesion dataset.**

| Setting | CBCT | | DeepLesion | |
|---|---|---|---|---|
| | SSIM | PSNR | SSIM | PSNR |
| S1 | 0.9303 ± 0.0812 | 34.28 ± 1.95 | 0.9305 ± 0.2433 | 32.85 ± 2.78 |
| S1 + S2 | 0.9527 ± 0.0534 | 35.86 ± 1.38 | 0.9396 ± 0.2153 | 34.71 ± 1.88 |
| S1 + S3 | 0.9579 ± 0.0572 | 36.35 ± 1.58 | 0.9427 ± 0.2272 | 35.02 ± 2.15 |
| S2 + S3 | 0.9531 ± 0.0423 | 36.21 ± 1.86 | 0.9404 ± 0.1955 | 34.88 ± 2.32 |
| S1 + S2 + S3 | **0.9634 ± 0.0295** | **36.91 ± 1.08** | **0.9502 ± 0.0872** | **35.93 ± 1.99** |
| Image domain | 0.9601 ± 0.0563 | 35.87 ± 1.56 | 0.9371 ± 0.1732 | 34.35 ± 2.30 |
| Dual-domain | **0.9634 ± 0.0295** | **36.91 ± 1.08** | **0.9502 ± 0.0872** | **35.93 ± 1.99** |

Best performance is highlighted in bold font.

streaks across images and severely impede the detection and diagnosis of disease. Metal artifact reduction (MAR) has hence of great importance in clinical practice for decades[48–51]. In this experiment, we evaluate the effectiveness of the proposed framework on MAR in CT images and hence, Radon transform is used for transformation between domains.

*Performance evaluation.* To demonstrate the advancement of our framework for MAR, we compare with other representative MAR methods such as the conventional linear interpolation (LI)[48] and normalized metal artifact reduction (NMAR)[49] methods, and learning-based RCN[50], CycleGAN[39], attention-based MAR[51] (AttenMAR), and DudoNet[28] on two datasets (with details provided in the "Methods" section). It is worth noting that our MAR model does not depend on complex pre-processing steps, such as pre-segmentation of the implant, and the model is easy to train

and re-implement. Since we have metal-free images as references, we adopt the conventional metrics PSNR and SSIM for quantitative assessment. From the experimental results provided in Table 3, we can observe that AttenMAR and the proposed model achieve much better performance than the others in terms of average SSIM and PSNR on both datasets. In comparison with AttenMAR, the performance gain of our model is mainly from the use of hierarchical consistency within and across domains.

*Effectiveness of dual-domain hierarchical consistency in MAR.* We have shown the effectiveness of hierarchical consistency and dual-domain on low-dose PET/CT reconstruction in the previous section. In this additional experiment, we validate the influence of hierarchical consistency within and across domains on MAR. We perform an ablation study and report the results in Table 4. We can see the impact of hierarchical consistency constraints from S1

**Table 5 Quantitative evaluation of our framework based on two representative backbones UNet and E2EVarNet using the in-house dataset for 4× and 8× acceleration rates.**

| Network | 4× | | 8× | |
|---|---|---|---|---|
| | SSIM | PSNR | SSIM | PSNR |
| UNet[53] | 0.9808 ± 0.0061 | 37.05 ± 1.89 | 0.9734 ± 0.0093 | 35.31 ± 2.23 |
| UNet (Ours) | **0.9866 ± 0.0041** | **39.53 ± 1.73** | **0.9782 ± 0.0069** | **36.88 ± 1.80** |
| E2EVarNet[54] | 0.9653 ± 0.0073 | 36.12 ± 1.64 | 0.9481 ± 0.0121 | 33.29 ± 1.41 |
| E2EVarNet (Ours) | **0.9703 ± 0.0069** | **37.00 ± 1.46** | **0.9561 ± 0.0116** | **34.13 ± 1.38** |
| DuDoRNet[20] | 0.9861 ± 0.0055 | 38.08 ± 2.21 | 0.9733 ± 0.0110 | 34.70 ± 2.10 |

Best performance is highlighted in bold font.

**Table 6 Ablation study of the proposed framework for MRI reconstruction based on the UNet backbone on the in-house dataset for 4× and 8× acceleration rates.**

| Setting | 4× | | 8× | |
|---|---|---|---|---|
| | SSIM | PSNR | SSIM | PSNR |
| S1 | 0.9808 ± 0.0061 | 37.05 ± 1.89 | 0.9734 ± 0.0093 | 35.31 ± 2.23 |
| S1 + S2 | 0.9862 ± 0.0042 | 39.13 ± 1.81 | 0.9762 ± 0.0085 | 35.75 ± 2.49 |
| S1 + S3 | 0.9862 ± 0.0036 | 39.48 ± 1.64 | 0.9781 ± 0.0069 | 36.04 ± 1.71 |
| S2 + S3 | 0.9863 ± 0.0038 | 39.50 ± 1.71 | 0.9781 ± 0.0062 | 35.87 ± 1.38 |
| S1 + S2 + S3 | **0.9866 ± 0.0041** | **39.53 ± 1.73** | **0.9782 ± 0.0069** | **36.88 ± 1.80** |
| Image domain | 0.9808 ± 0.0061 | 37.05 ± 1.89 | 0.9734 ± 0.0093 | 35.31 ± 2.23 |
| Dual-domain | **0.9866 ± 0.0041** | **39.53 ± 1.73** | **0.9782 ± 0.0069** | **36.88 ± 1.80** |

Best performance is highlighted in bold font.

to S3 on both datasets. The multi-stages scheme not only improves the mean of different assessment metrics but also the individual standard deviation, which indicates better robustness for different test samples. Besides, the use of a dual-domain scheme allows greater performance improvement than a single-domain scheme in terms of SSIM, PSNR, and NRMSE.

**Fast MRI reconstruction**. MRI is a commonly used non-invasive and radiation-free medical imaging technique. Despite its advantages in high spatial resolution and multi contrasts, a major limitation of MRI is the slow acquisition speed, since multiple times of radiofrequency (RF) pulses are required to fill in the $k$-space for encoding spatial-frequency information, and also each contrast has to be scanned separately. Consequently, such lengthy acquisitions can lead to patient discomfort and severe motion artifacts in the acquired images[52]. Meanwhile, this also limits the availability of scanners. Therefore, the development of fast MRI reconstruction algorithms can greatly improve the efficiency of data acquisition and subsequent diagnosis. In this experiment, we validate our framework for fast MRI reconstruction and use Fourier transform as the transform function $F$.

*Performance evaluation*. For the sake of demonstrating the effectiveness of dual-domain and hierarchical consistency, we evaluate the performance of our framework using two backbones, namely UNet[53] and E2EVarNet[54] as the generative functions. In particular, UNet acts as $G_t^I, G_t^A$ in both domains and shares the same weights. Data-consistency mapping[55] is utilized as $G_s^I, G_s^A$. The same setting is implemented for the E2EVarNet backbone. It is worth noting that more sophisticated generative functions can be used in our framework for possibly better performance. We evaluate the proposed reconstruction model using an in-house MRI dataset containing 62 T2-weighted (T2w) MR images. We compare reconstructed images by our model with the

corresponding baseline methods, for the acceleration rates of 4× and 8×. Quantitative evaluation is provided in Table 5. In addition, we also compare with the state-of-the-art MRI reconstruction method, i.e., DuDoRNet[20]. Due to hardware restriction, the number of recurrent blocks is set as 2 for fair comparison, and the optimization is performed for 1000 epochs. As can be observed, compared to both representative baseline methods, our proposed model provides better performance in all the evaluation metrics for both acceleration rates.

*Effectiveness of dual-domain hierarchical consistency in MRI Reconstruction*. By taking the native dual-domain representation of MRI data into account, our framework can exploit patterns in both domains and regularize the optimization in a structured manner. To demonstrate the effectiveness of the introduced dual-domain hierarchical consistency constraints, we conduct an ablation study and summarize the quantitative analysis in Table 6. As we can see, the hierarchical consistency constraints improve the performance of MRI reconstruction stepwise which coincides with the aforementioned results for the PET/CT reconstruction and MAR tasks. Besides, it is shown that the utilization of dual-domain information brings a noticeable improvement in reconstruction performance for both acceleration factors than using solely single-domain data.

**PET-CT synthesis**. Both PET-to-CT and CT-to-PET syntheses have great potential in clinical applications. In routine clinical practice, CT is used for anatomical localization for PET. Synthesis of CT from PET can avoid additional radiation caused by CT, which is of great importance for reducing radiation dose. From another perspective, since PET is more expensive than CT and is not as easily available as CT, synthesis of PET from CT is in fact also of great practical interest. In this experiment, we use PET-to-CT and CT-to-PET syntheses as case studies to evaluate and analyze the effectiveness of our proposed framework on synthesis tasks.

**Table 7 Quantitative comparison with other state-of-the-art methods for PET-CT synthesis based on the in-house dataset.**

| Tasks | PET-to-CT | | CT-to-PET | | | |
|---|---|---|---|---|---|---|
| Method | SSIM | SSIM | SSIM | PSNR | SUV$_{mean}$ | SUV$_{max}$ |
| UNet[44] | 0.9581 ± 0.0427 | 35.10 ± 1.06 | 0.9160 ± 0.0357 | 33.52 ± 1.05 | 9.02 ± 4.22 | −7.48 ± 6.86 |
| RUNet[45] | 0.9569 ± 0.0484 | 35.54 ± 1.02 | 0.9491 ± 0.0390 | 34.82 ± 1.62 | 8.12 ± 2.74 | −9.62 ± 1.07 |
| p2pGAN[46] | 0.9666 ± 0.0200 | 35.18 ± 0.98 | 0.9391 ± 0.0484 | 35.04 ± 1.67 | 9.52 ± 3.19 | −9.36 ± 1.90 |
| CycleGAN[40] | 0.9646 ± 0.0197 | 35.49 ± 1.04 | 0.9581 ± 0.0402 | 35.70 ± 2.01 | 8.86 ± 6.43 | −8.06 ± 2.81 |
| MedGAN[38] | 0.9706 ± 0.0111 | 35.70 ± 0.64 | 0.9091 ± 0.0658 | 35.83 ± 2.77 | 7.36 ± 7.58 | −5.04 ± 2.07 |
| Ours | **0.9843 ± 0.0123** | **38.33 ± 0.55** | **0.9658 ± 0.0436** | **36.76 ± 1.25** | **4.98 ± 3.07** | **3.27 ± 4.46** |

Best performance is highlighted in bold font.

**Table 8 Ablation study of the proposed framework for PET-CT synthesis on in-house dataset.**

| Setting | PET-to-CT | | CT-to-PET | |
|---|---|---|---|---|
| | SSIM | PSNR | SSIM | PSNR |
| S1 | 0.9569 ± 0.0484 | 35.54 ± 1.02 | 0.9491 ± 0.0390 | 34.82 ± 1.62 |
| S1 + S2 | 0.9810 ± 0.0319 | 37.30 ± 1.21 | 0.9612 ± 0.0312 | 36.02 ± 1.27 |
| S1 + S3 | 0.9573 ± 0.0482 | 37.15 ± 1.76 | 0.9486 ± 0.0582 | 35.66 ± 1.89 |
| S2 + S3 | 0.9828 ± 0.0327 | 38.01 ± 1.11 | 0.9384 ± 0.0598 | 36.21 ± 1.58 |
| S1 + S2 + S3 | **0.9843 ± 0.0123** | **38.33 ± 0.55** | **0.9658 ± 0.0436** | **36.76 ± 1.25** |
| Image domain | 0.9783 ± 0.0155 | 36.24 ± 1.01 | 0.9379 ± 0.0759 | 34.84 ± 1.96 |
| Dual-domain | **0.9843 ± 0.0123** | **38.33 ± 0.55** | **0.9658 ± 0.0436** | **36.76 ± 1.25** |

Best performance is highlighted in bold font.

*Performance evaluation.* In this experiment, our network has the same structure as the one used for low-dose PET/CT reconstruction. We compare our synthesis model with the representative approaches, including UNet[44], RUNet[45], p2pGAN[46], CycleGAN[40], and MedGAN[38]. These networks are widely used for medical image synthesis, especially for PET-CT synthesis. Note that only Cycle-GAN and our framework can jointly learn the two tasks, i.e., PET-to-CT synthesis and CT-to-PET synthesis. Others are single-direction synthesis models and are thus trained for these two synthesis tasks independently. The quantitative results of all the studied models are summarized in Table 7. For the PET-to-CT task, our framework achieves SSIM up to 0.9843 and outperforms the other methods by a large margin. With respect to the CT-to-PET task, the proposed network also shows superior performance than the other models. To evaluate the SUV bias of reconstructed PET images in a CT-to-PET reconstruction task, we demonstrate the performance of each method in Table 7, indicating the superiority and feasibility of our framework. To better visualize the performance improvement, we show perceptual evaluation in Fig. 3. The PET-to-CT images are demonstrated in the left panel, and the CT-to-PET results are shown in the right panel. For each task, we illustrate six locations of the human brain. We can clearly see that, in comparison to other methods, the synthesized CT and PET images by the proposed model are more consistent with the GT images, which coincides with the above quantitative assessment. Our results indicate that, by resorting to dual-domain cycle consistency, the proposed generative framework achieves promising performance for synthesis tasks. The noise power spectrum comparisons presented in Fig. 5 provide further evidence for the superior performance of our method. In both PET-to-CT and CT-to-PET synthesis tasks, our method exhibits the lowest noise power across spatial frequency. This indicates the great advantage of our dual-domain and hierarchical consistency learning approach.

*Effectiveness of dual-domain hierarchical consistency in PET-CT synthesis.* To better understand the effectiveness of each stage for synthesis tasks, we perform an ablation study on PET-CT synthesis and provide experimental results in Table 8. When comparing the performance obtained by S1 + S2 against S1, we can clearly see the impact of inter-domain consistency on the synthesis performance. When integrating the cycle consistency S3 into S1 + S2, we can see further improvement for both cases, which indicates the importance of cycle consistency within and across domains. From quantitative analysis, we can conclude that the multi-stage consistency constraints are well-designed for our framework and each stage provides different kinds of supervision to improve overall performance in a complementary way. Besides, we also conduct experiments to evaluate the dual-domain scheme for PET-CT synthesis tasks. Particularly, we compare the performance with and without using the sinogram domain information and list the quantitative results in Table 8. We can see great performance improvement induced by the sinogram domain information in terms of average SSIM, PSNR, and NRMSE for both PET-to-CT and CT-to-PET synthesis tasks.

## Discussion

In this paper, we propose a generalized dual-domain framework for medical image reconstruction and synthesis based on hierarchical consistency constraints within and across domains. In particular, the dual domain can be interpreted as an image domain and another domain of interest, such as an image acquisition domain. Different from the conventional CycleGAN framework, where cycle consistency is performed between the source and target images using an unsupervised learning scheme in a single modality, e.g., the image domain, our proposed dual-domain-based generative framework adopts the principle of hierarchical consistency in dual domains based on supervised learning. Leveraging dual domains *not only* coincides with the inherent characteristics of medical imaging *but also* allows better exploitation of the underlying patterns in both acquisition and image domains. Without loss of generality, by involving four

generative functions between the source and target images in dual domains, bi-directional mappings across images and domains are enabled, although, for certain tasks, some of the generative functions are not necessary to be learned. More importantly, unlike most of the existing dual-domain-based generative methods which either adopt sequentially cascaded or parallel-connected sub-networks for processing the individual domain patterns, we explicitly impose hierarchical consistency, including intra-domain consistency, inter-domain consistency, and cycle consistency. These hierarchical consistency constraints are stepwise integrated into three stages of generative framework during the training, to achieve a multi-level similarity match in a stabilized and structured way. In extensive experiments, multiple representative generative tasks are investigated, and the proposed generative framework achieves superior performance compared to the corresponding state-of-the-art methods in different applications. Furthermore, we have performed comprehensive analysis and in-depth ablation study from different perspectives to evaluate the effectiveness of the dual-domain and hierarchical consistency design in several representative generative tasks.

We first carried out experiments to evaluate our framework for low-dose PET and low-dose CT reconstruction. We collected 70 standard-dose PET volumes with a total of 1540 slices and also 8 standard-dose CT volumes with 5326 slices in total. Following the standard simulation procedure, we generated corresponding low-dose counterparts. By using hierarchical dual-domain constraints, the proposed reconstruction model obtains great performance gain compared to representative reconstruction methods.

Furthermore, we have also evaluated our framework for metal artifact reduction in Cone-beam CT (CBCT). We conducted experiments on two datasets. The first one contains 100 CT volumes of teeth from local hospitals with 5500 selected slices in total, and the second one is a subset of the public dataset DeepLesion which contains 4118 slices. Compared to competitive approaches such as AttenMAR[51] and RCN[50], our proposed framework exhibits remarkable improvement in terms of average PSNR and SSIM on both datasets.

To further validate the effectiveness of our proposed framework, we carried out experiments for MRI reconstruction. The development of an MRI reconstruction algorithm is of clinical importance since it can improve image quality by alleviating severe aliasing effects due to $k$-space subsampling. In the experiment, an in-house dataset which consists of 24-coil T2w MR brain images of 62 subjects is employed. We construct our reconstruction model based on two representative backbones, namely UNet[53] and E2EVarNet[54]. Experiments show that the proposed framework obtains noticeable performance gain on both backbones for 4× and 8× acceleration rates.

Besides experiments for reconstruction tasks, we perform experimental evaluation also for the task of PET-CT synthesis based on an in-house dataset containing 65 paired PET/CT brain volumes. In comparison with existing state-of-the-art methods for image synthesis such as MedGAN[38] and p2pGAN[46], our proposed framework achieves great quantitative improvement, which coincides with the qualitative performance in visual perception.

In summary, from the above evaluations for different reconstruction and synthesis tasks, we can see that, one can always use a UNet-shaped network as the baseline backbone for the individual generative functions $G$, and separately designed network structure for different applications can further facilitate the model performance. Moreover, although different generative functions can be selected for different applications, all models share the same dual-domain framework with hierarchical consistency constraints and achieve remarkable performance improvement in their respective applications. Therefore, we can conclude that the proposed generative framework is general for medical image reconstruction and synthesis.

## Methods
### Datasets

*Low-dose PET/CT reconstruction.* For the low-dose reconstruction task, 70 low-dose PET images were simulated from the corresponding standard-dose counterparts according to the sampling principle of PET. Specifically, we randomly generated low-dose sinograms by downsampling the standard-dose ones and used the OSEM algorithm to get paired low-dose and standard-dose PET images. We cropped and resampled the paired volumes to the resolution of 128 × 128 with a pixel size of 2.344 × 2.344 mm$^2$. In the experiments, 1078 2D slices were randomly chosen as the training set, 154 slices as the validation set, and 308 samples as the testing set. For low-dose CT reconstruction, we collected eight standard-dose CT volumes and simulated the low-dose CT volumes by decreasing the operating current (mA) and the number of projections. We cropped and resampled CT volumes to the same resolution, and used slices in the axial view to generate the 2D dataset with 5326 samples in total. The resolution of low-dose and standard-dose CT images is 512 × 512 with a pixel size of 0.820 × 0.820 mm$^2$. We randomly chose 3897 samples as the training set, 585 samples as the validation set, and the rest 844 samples as the testing set.

*Metal artifact reduction.* We have validated the proposed dual-domain cycle-consistent MAR network on two datasets, one in-house Cone-Beam CT (CBCT) dataset of teeth images and one public dataset (a subset of DeepLesion[56]). For the first dataset, we collected 100 high-quality teeth CBCT volumes from local hospitals. All the CT volumes have the same resolution of 1 × 1 × 1 mm$^3$ and the same size of 400 × 400 in the transverse view. All the individual teeth were annotated by experienced dentists. We randomly selected 5500 slices from the collected metal-free volumes, in which 4400 slices were used for training and the rest 1100 for testing. We followed the experiment setup in ref. [57] to simulate the metal-affected images, where 1–4 teeth were selected as the implant metals based on the segmentation annotation in each selected slice for simulating metal artifacts. For the second dataset, we selected a subset of DeepLesion as suggested by Liao et al.[14]. 3918 metal-free CT slices in conjunction with 90 metal masks were employed to generate metal-affected images for training, and other 200 metal-free CT images with 10 metal masks were used for testing.

*Fast MRI reconstruction.* An in-house dataset consisting of 62 multi-coil MR brain images was used to evaluate our proposed framework for MRI reconstruction. The dataset contains 24-coil $k$-space data of 62 subjects scanned with a 3T MR scanner using a T2w pulse sequence (TR = 4226 ms, TE = 104.8 ms). Each image has a spatial resolution of 0.7 × 0.7 × 5 mm$^3$. To augment the training dataset, rigid transform with a random rotation within ± 10° and random translation in the range of ±15 mm was applied to the data. For each subject, we normalized the image based on intensity and extracted the axial slices for both contrasts. The 24-channel complex-valued images were cropped to 320 × 320 to remove the background area and the Fourier transform was performed to obtain the corresponding $k$-space data. A Cartesian Gaussian random under-sampling pattern was applied to achieve the acceleration rate of 4× and 8×, where 6 center lines were always

acquired. In such a way, we extracted 1248 slices from the 62 subjects and randomly split them into 503, 241, and 504 for training, validation, and testing, respectively.

*PET-CT synthesis.* For PET-CT synthesis, we used 65 paired PET and CT brain images acquired from the uEXPLORE PET/CT system. We cropped and resampled these paired volumes to the resolution of $128 \times 128$ with a pixel size of $2.344 \times 2.344$ mm$^2$. We selected the middle 20 axial slices to aggregate a total of 1300 samples. 1000 samples were randomly chosen as the training set, 60 samples were set as the validation set, and the rest 240 samples were used for testing.

**Generative framework.** As shown in Fig. 1, our proposed framework consists of four generative functions, namely the target-to-source $G_s^I$ and the source-to-target $G_t^I$ in the image domain and $G_s^A$, $G_t^A$ in the second domain such as the sinogram domain. Depending on the applications, the generative functions can be as complex as highly nonlinear mappings such as convolutional neural networks, or as simple as linear mappings such as a fully connected layer or a sampling map. Each function aims to build up a mapping between the source and target images in the corresponding domain. In conjunction with the forward and backward transforms $F$ and $F^{-1}$ such as the Radon transform for CT and PET while the Fourier transform for MRI, one can construct a dual-domain and bidirectional framework as shown in Fig. 1b. Thanks to the architecture of the proposed framework, one can exploit and aggregate patterns in the individual domain, and also build up multi-stage bidirectional consistency constraints. By resorting to the hierarchical bidirectional constraints, one can strengthen the coupling of different representations and obtain a more robust solution to the inverse problem. In particular, the hierarchical bidirectional consistency constraints consist of three stages: Stage 1 performs intra-domain consistency by building up relations and constraints between the source and target images in each domain independently; Stage 2 aims for constructing connections across domains to preserve inter-domain consistency; Stage 3 integrates bidirectional cycle consistency within and across domains into the framework to achieve hierarchical consistency. Detailed description of each stage is given below.

**Algorithm 1. Multi-stage training strategy**

**Input** : Training set $\{x_s, x_t\}_1^{num}$

**Require** : Networks: $G_t^I$ with trainable parameters $\theta_1$, $G_s^I$ with trainable parameters $\theta_2$, $G_t^A$ with trainable parameters $\theta_3$,
$G_s^A$ with trainable parameters $\theta_4$; Hyper-parameters: $\lambda_1$, $\lambda_2$, $\lambda_3$, $\lambda_4$, $\xi_1$, $\xi_2$, $\xi_3$, $\xi_4$

1 Randomly initialize $\theta_1$, $\theta_2$, $\theta_3$, $\theta_4$
2 **while** *not done* **do**
3 Update $\theta_1$, $\theta_2$, $\theta_3$, $\theta_4$ according to Equation (1)
4 **end**
5 **while** *not done* **do**
6 Fix $\theta_1$ and update $\theta_3$ according to Equation (2)
7 Fix $\theta_3$ and update $\theta_1$ according to Equation (3)
8 Fix $\theta_2$ and update $\theta_4$ according to Equation (4)
9 Fix $\theta_4$ and update $\theta_2$ according to Equation (5)
10 **end**
11 Fix $\theta_3$ and $\theta_4$
12 **while** *not done* **do**
13 Fix $\theta_1$ and update $\theta_2$ according to Equation (6)
14 Fix $\theta_2$ and update $\theta_1$ according to Equation (7)
15 **end**

**Output** : Well-trained network $G_t^I$ with parameters $\theta_1$ and $G_s^I$ with parameters $\theta_2$

*Stage 1: Intra-domain consistency.* As the cornerstone of hierarchical consistency, intra-domain consistency is achieved by training the generative functions $G_s^I, G_s^A, G_t^I, G_t^A$ independently in a supervised manner. Specifically, we perform bidirectional training between source and target images in each domain to set up the shortest local consistency and pave the way for inter-domain consistency in the second stage. Given the paired source image $x_s$ and target image $x_t$ such as paired CT and PET or artifact-contaminated and artifact-free CT images, the individual loss functions of $G_s^I, G_s^A, G_t^I, G_t^A$ are formulated as follows:

$$
\begin{aligned}
\mathcal{L}_t^I &= \mathbb{E}_{x_s,x_t} ||G_t^I(x_s) - x_t||_1, \\
\mathcal{L}_s^I &= \mathbb{E}_{x_s,x_t} ||G_s^I(x_t) - x_s||_1, \\
\mathcal{L}_t^A &= \mathbb{E}_{x_s,x_t} ||G_t^A(F(x_s)) - F(x_t)||_1, \\
\mathcal{L}_s^A &= \mathbb{E}_{x_s,x_t} ||G_s^A(F(x_t)) - F(x_s)||_1.
\end{aligned}
\tag{1}
$$

*Stage 2: Inter-domain consistency.* Stage 2 is designed to construct and strengthen the coupling of dual-domain information for inter-domain consistency by integrating additional cross-domain constraints. In particular, the bidirectional consistency, namely from source to target and target to source, is performed separately. For the source-to-target direction, we alternately train $G_t^I$ and $G_t^A$ by the loss defined in Eqs. (2) and (3), respectively. When training $G_t^I$, we freeze the parameters of $G_t^A$; and when training $G_t^A$, we freeze $G_t^I$. Following such a training strategy, we can achieve cross-domain consistency for the source-to-target direction efficiently.

$$
\begin{aligned}
\mathcal{L}_t^I &= \mathbb{E}_{x_s,x_t} ||G_t^I(x_s) - x_t||_1 \\
&+ \lambda_1 \mathbb{E}_{x_s} ||G_t^I(x_s) - F^{-1}(G_t^A(F(x_s)))||_1,
\end{aligned}
\tag{2}
$$

$$
\begin{aligned}
\mathcal{L}_t^A &= \mathbb{E}_{x_s,x_t} ||G_t^A(F(x_s)) - F(x_t)||_1 \\
&+ \lambda_2 \mathbb{E}_{x_s} ||G_t^A(F(x_s)) - F(G_t^I(x_s))||_1,
\end{aligned}
\tag{3}
$$

Similarly, based on the loss functions in Eqs. (4) and (5), we can train $G_s^I$ and $G_s^A$ alternatively and preserve the dual-domain consistency for the target-to-source direction.

$$
\begin{aligned}
\mathcal{L}_s^I &= \mathbb{E}_{x_s,x_t} ||G_s^I(x_t) - x_s||_1 \\
&+ \lambda_3 \mathbb{E}_{x_t} ||G_s^I(x_t) - F^{-1}(G_s^A(F(x_t)))||_1,
\end{aligned}
\tag{4}
$$

$$\mathcal{L}_s^A = \mathbb{E}_{x_s, x_t} ||G_s^A(F(x_t)) - F(x_s)||_1 \\ + \lambda_4 \mathbb{E}_{x_t} ||G_s^A(F(x_t)) - F(G_s^I(x_t))||_1, \quad (5)$$

*Stage 3: Cycle consistency.* In Stage 2, we have built up the consistency from source to target and target to source. In Stage 3, we intend to extend the consistency constraints by involving cycle consistency within and across domains. The cycle consistency has two directions, namely from source to source in anti-clockwise direction and from target to target in clockwise direction. The source-to-source direction is to fine-tune $G_t^I$ by freezing the parameters of the other functions, while the target-to-target is to fine-tune $G_s^I$. Each of these two directions contains cycle consistencies in dual domains. For example, for the source-to-source direction, the source image goes through $G_t^I$ and $G_s^I$ to complete the cycle in the image domain. For the dual-domain cycle, in conjunction with the pre-trained $G_s^A$ and $G_t^A$ whose parameters are frozen, the anti-clockwise cycle across domains preserves the self-similarity of the source image. The loss function for updating $G_t^I$ based on the source-to-source cycle is described as

$$\mathcal{L}_t = \mathbb{E}_{x_s, x_t} ||G_t^I(x_s) - x_t||_1 \\ + \xi_1 \mathbb{E}_{x_s} ||G_s^I(G_t^I(x_s)) - x_s||_1 \\ + \xi_2 \mathbb{E}_{x_s} ||F^{-1}(G_s^A(F(G_t^I(x_s)))) - x_s||_1, \quad (6)$$

The first term is the primary supervision as defined in Stage 1. The second and the third terms are, respectively, the image-domain cycle consistency loss and the cross-domain cycle consistency loss. $\xi_1$ and $\xi_2$ are hyper-parameters to balance the importance of the two cycle-consistency losses.

Similarly, the target-to-target direction for updating $G_s^I$ follows the same scheme as above. The loss function for the clockwise direction is expressed in Eq. (7).

$$\mathcal{L}_s = \mathbb{E}_{x_s, x_t} ||G_s^I(x_t) - x_s||_1 \\ + \xi_3 \mathbb{E}_{x_t} ||G_t^I(G_s^I(x_t)) - x_t||_1 \\ + \xi_4 \mathbb{E}_{x_t} ||F^{-1}(G_t^A(F(G_s^I(x_t)))) - x_t||_1, \quad (7)$$

The hierarchical consistency constraints and the corresponding training strategy are summarized in Algorithm 1. It is worthy to mention that in the testing phase, we only perform $G_t^I$ or $G_s^I$ to obtain the network output although both directions have been trained.

**Statistics and reproducibility**. Our proposed framework was implemented on the PyTorch platform, and all the experiments were conducted on the NVIDIA GeForce GTX A100 with 40 GB RAM and the Intel(R) Xeon(R) Silver 4214R CPU.

For low-dose PET/CT reconstruction and PET-CT synthesis, we used RUNet[45] as the generative functions, i.e., $G_s^I$ and $G_t^I$, and adopted the Fully Connected Network (FCN)[58] as $G_s^A$ and $G_t^A$. We set the weights $\lambda_1$–$\lambda_4$ and $\xi_1$–$\xi_4$ as 0.5. Adam optimizer was employed to train four networks in all stages, and the learning rate of three stages was set as 0.005, 0.002, and 0.002, respectively. For low-dose PET reconstruction, the training time of our method is about 21 h, and other comparison methods took about 13 h (RUNet), 17 h (p2pGAN), 17.5 h (MedGAN), 16.5 h (CycleGAN), and 19 h (iBP-Net) for training. For low-dose CT reconstruction, the training time of our method is about 71 h, and other comparison methods took about 36.5 h (RUNet), 49 h (p2pGAN), 55 h (MedGAN), 52.5 h (CycleGAN), and 63.5 h (iBP-Net) for training. For PET-CT synthesis, the training time of our method is about 22.5 h, and other comparison methods took about 25.5 h (U-Net), 25.5 h (RUNet), 30.5 h (p2pGAN), 17 h (MedGAN), and 35 h (CycleGAN) for training.

For MAR, we employed four identical RUNet (initial filters of 8, 16, and 32, respectively) as the generative functions for both domains. For the CBCT dataset, the minimum and maximum CT densities were respectively clipped to −1000 HU and 2000 HU. All the images were normalized based on the attenuation coefficients ($\mu_{H_2O} = 0.192$) as the network input. For the Deeplesion dataset, we follow the preprocessing in ref. [50]. The network was optimized by the Adam optimizer with a learning rate of 0.0001 for 20 epochs and the mini-batch size was set as 8 and 6 for CBCT and Deeplesion, respectively. The weights of the spatial domain loss and projection loss were set as 10 and 0.1, respectively, because the intensity values in the projection domain are usually large. The weighting parameters $\lambda_1$–$\lambda_4$ for each stage were chosen as 0.2 and $\xi_1$–$\xi_4$ were set as 0.1. The experiments are performed on a shared server. The training time of our method is about 18.5 h for the CBCT dataset and other compared methods are about 14.5 h (CycleGAN), 11 h (RCN), 8 h (AttentionMAR), and 16 h (DuDoNet). The training time of our method on the Deeplesion dataset is about 13 h and other compared methods are about 16 h (CycleGAN), 14 h (RCN), 13 h (AttentionMAR), and 18 h (DuDoNet).

For fast MRI reconstruction, the proposed framework is evaluated on the backbones of UNet and E2EVarNet. The constructed models were trained until convergence within 100 epochs with the learning rate of $2 \times 10^{-4}$ using the Adam optimizer[59]. The designed model for MRI reconstruction consists of only two generative functions, namely $G_t^I, G_t^A$ from the source to the target direction, the loss terms associated with the coefficients of $\lambda_3$, $\lambda_4$, $\xi_1$, and $\xi_3$ are not applicable. The values of $\lambda_1$ and $\lambda_2$ were both chosen as 0.5. The weights $\xi_2$ and $\xi_4$ were empirically selected as 0.5 by simple grid searching on the validation set. The output of the reconstruction network in the image domain is used during inference. The training time of our method is about 1.5 h with UNet as the backbone, and 6.5 h with E2EVarNet as the backbone, while the comparison method DuDoRNet took 15.5 h for training.

**Data availability**

The authors declare that partial data will be released to support the results of this study (https://github.com/ZhangJD-ong/Medical-image-reconstruction-and-synthesis), with permission from respective data centers. The full datasets are protected because of privacy issues and regular policies in hospitals. All relevant data supporting the findings of this study are available from the corresponding author upon reasonable request. Any data use will be restricted to non-commercial research purposes.

**Code availability**

All custom code used in this work, including that used to train and test the framework, can be obtained from the following publicly accessible GitHub page: https://github.com/ZhangJD-ong/Medical-image-reconstruction-and-synthesis.

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

## Acknowledgements

This work was supported in part by the National Natural Science Foundation of China (grant number 62131015), the Science and Technology Commission of Shanghai Municipality (STCSM) (grant number 21010502600), and The Key R&D Program of Guangdong Province, China (grant number 2021B0101420006).

## Author contributions

Z.C. and D.S. participated in the study concept and design. J.Z., J.Y., and Y.H. participated in the study conduct and data analysis. K.S. took the lead on manuscript writing, along with J.Z., J.Y., Y.H., and F.G. Y.G., X.Z., and D.S. revised the manuscript. The whole project was managed by D.S. All authors read and proved the final manuscript.

## Competing interests

The authors declare no competing interests.
