## [Peer Review File · Communications Engineering]

Reviewers' comments:

Reviewer #1 (Remarks to the Author):

This paper provides a general dual-domain learning for different tasks, including CT, PET and MRI reconstruction, metal artifacts reduction. But the reviewer has following main concerns for this study.

1. The dual domain learning is a very common strategy of reconstruction, synthesis and MAR etc. I just wonder the novelty of this study, it seems this study mainly focus on the application of dual domain learning method.
2. In respect to the proposed method, including cycle consistency, inter-domain consistency and intra-domain consistency, the fundamental principle is very similar to the CycleGAN. What the difference between these two methods.
3. In terms of demonstrating the advantages of the proposed with reconstruction results, only the error to the ground truth is shown and that is not sufficient. This reviewer suggests the authors should proposed noise power spectrum comparison.
4. The comparison methods with respect to reconstruction and MAR are too old. The authors should add advanced reconstruction methods to highlight the performance of proposed algorithm.
5. In Table 1, the metrics of PSNR and RMSE are the same meaning, the authors can give one of them.
6. In respect to the table 2, the authors try to highlight the dual domain advantages over single domain, in fact, there are some great works have investigated. Therefore, this reviewer still concerns the originality of this paper.

Reviewer #2 (Remarks to the Author):

The authors proposed a generalized dual-domain generative framework with hierarchical-consistency for medical image reconstruction and synthesis. The performance was demonstrated by low-dose PET/CT reconstruction, metal artifact reduction, fast MRI reconstruction, and PET-CT synthesis. Overall, the paper is well organized, and it can be further improved from the following aspects.

1. In recent two years, many dual-domain deep learning based networks were reported for medical image reconstruction and synthesis. The authors may need a deep survey on this topic in the introduction to enhance the background of this paper.
2. For all the experimental results, in Tables 1-8, both PSNR and NRMSE are used. However, PSNR are identical to NRMSE up to a log operation and a constant bias. The authors may consider keep one of them to remove the redundancy.
3. Regarding the ablation study, in Tables 2, 4, 6 and 8, to demonstrate that all the components are necessary, experiments should be performed for all the settings without the testing component. Hence, the authors may consider the settings "S2+S3" and "S1+S3".
4. In all the tables, the values of SSIM are close to 1.0, and almost all the values are greater than 0.95. The differences between the proposed method and the competing methods are very small. Noting that the default parameters in the SSIM function were designed for nature images with a pixel value range of 0-255. Because the original medical image pixel values have different ranges, the default parameters in SSIM cannot be directly applied for medical images. What parameters are employed to compute SSIM for each of the experiment?

5. For a deep learning based method, network training is time-consuming. For all the deep learning based methods in each experiment, it is better to clarify the computational cost to train the proposed network framework and the related competing networks.
6. Regarding the application of low-dose PET/CT reconstruction, all the competing methods listed in Table 1 only use image-domain information. Since the proposed method is in dual-domain, at least one state-of-the-art dual-domain image reconstruction approach should be compared. For example:
Wu et al, DRONE: Dual-domain residual-based optimization network for sparse-view CT reconstruction, IEEE-TMI, 40(11):3002-3014, 2021.
Jiao et al, A dual-domain CNN-based network for CT reconstruction, IEEE Access, 9:71091-71103, 2021.
Zhou et al, DuDoUFNet: Dual-domain under-to-fully-complete progressive restoration network for simultaneous metal artifact reduction and low-dose CT reconstruction, IEEE-TMI, 41(12): 3587-3599, 2022.
7. Regarding the application of metal artifact reduction, at least one state-of-the-art dual-domain approach should be compared. For example, references 18 and 22 cited in this paper.
8. Regarding the application of fast MRI reconstruction, at least one state-of-the-art dual-domain approach should be compared. For example, references 19 and 21 cited in this paper.

Reviewer #3 (Remarks to the Author):

This paper presents a novel generalized dual-domain generative framework with hierarchical-consistency for medical image reconstruction and synthesis. Extensive experimental results demonstrate the effectiveness of the proposed method.

Strengths:

The motivation is clear. The paper is well-written and easy to follow.

Extensive experimental results are conducted to support the effectiveness of the method.

As the authors claimed, code will be published and some of the source datasets will also be released, which will facilitate the research in this area.

Weakness:

Just curious, in PET/CT reconstruction and PET-CT synthesis a fully connected network is employed for G_s^A and G_t^A , but in MAR, RU-Net is employed, and in fast MRI, E2EVarNet is employed. Are there any suggestions for the deployment of different architectures for different tasks? As a generalized framework, the method should work robustly no matter which network architectures are employed. The authors are encouraged to conduct experiments on one of the tasks by employing different network architectures as baseline networks.

Seems that the same metric is used in all the experimental results. A more thorough or clinically related metric may be considered for further evaluation on one of the dataset, for example (SUVs) for PET [Ref1, Ref2] or radiologist report for the synthesized images.

Ref1. Low-count whole-body PET/MRI restoration: an evaluation of dose reduction spectrum and five state-of-the-art artificial intelligence models

Ref2. Deep learning-assisted ultra-fast/low-dose whole-body PET/CT imaging

Author's Response to the Reviewers of Paper COMMS-23-0013

Title: A Generalized Dual-Domain Generative Framework with Hierarchical-Consistency for Medical Image Reconstruction and Synthesis

Paper No: COMMS-23-0013

By: Jiadong Zhang, Kaicong Sun, Junwei Yang, Yan Hu, Yuning Gu, Zhiming Cui, Xiaopeng Zong, Fei Gao, Dinggang Shen.

Dear Editor and Reviewers,

Thank you very much for your review work on our manuscript, which has guided us to improve the quality of our manuscript significantly. We have revised our manuscript according to your constructive comments point by point and the details are listed below.

Yours sincerely,

Dinggang Shen

Professor, Ph. D.

School of Biomedical Engineering,

ShanghaiTech University,

393 Middle Huaxia Road, Pudong, Shanghai, 2012102, China

E-mail: dgshen@shanghaitech.edu.cn

Reply to Reviewer 1

Paper No: COMMS-23-0013

By: Jiadong Zhang, Kaicong Sun, Junwei Yang, Yan Hu, Yuning Gu, Zhiming Cui, Xiaopeng Zong, Fei Gao, Dinggang Shen.

Dear Reviewer,

Thank you very much for your affirmation and constructive comments on our paper. Your suggestions have helped us a lot to improve the quality of our manuscript. Based on your comments and suggestions, we have made the following revisions.

[Comment 1]:

The dual domain learning is a very common strategy of reconstruction, synthesis and MAR etc. I just wonder the novelty of this study, it seems this study mainly focus on the application of dual domain learning method.

[Author's answer and modification]:

Thanks for your comment. We acknowledge that there are many dual-domain learning works for reconstruction tasks. However, most of these methods use dual-domain images as inputs of the cascaded dual-domain networks, which indeed involve the dual-domain knowledge but cannot guarantee the dual-domain consistency across two domains. To address this issue, we propose the dual-domain cycle-consistent generative framework by concerning dual-domain hierarchical consistency which intends to better regularize the potential solution space of medical image reconstruction or synthesis. Compared with other dual-domain works, to our best knowledge, the proposed framework is the first work that explicitly considers hierarchical inter- and intra-domain consistency constraints for medical synthesis and reconstruction which can better explore the latent physical relationships in medical images and hence achieve superior reconstruction performance.

[Comment 2]:

In respect to the proposed method, including cycle consistency, inter-domain consistency and intra-domain consistency, the fundamental principle is very similar to the CycleGAN. What the difference between these two methods.

[Author's answer and modification]:

Thanks for your comment. It's true that the cycle-consistent concept is proposed by CycleGAN. Based on the initial idea of cycle-consistent, we further extend the concept in our dual-domain framework. We not only apply the cycle-consistency constraint in

the image domain cycle, more importantly, we propose to apply hierarchical cycle-consistency constraint in the cross domain cycle. To our best knowledge, it's the first work to explore the dual-domain consistency in a unify framework, which is the main difference between the two methods.

[Comment 3]:

In terms of demonstrating the advantages of the proposed with reconstruction results, only the error to the ground truth is shown and that is not sufficient. This reviewer suggests the authors should proposed noise power spectrum comparison.

[Author's answer and modification]:

Thanks for your constructive comment. As the reviewer suggested, in the revised version, we have added the plot of noise power spectrum for all the studied medical applications as shown in Fig. 3 and Fig. 5. The error map between the synthesized image and the GT image is used as the noise map for NPS calculation. Referring to Dobbins III et. al [1], we use nonoverlapping region of interest with size of 64x64 for NPS calculation for low-dose PET/CT, Fast MRI reconstruction, and PET-CT synthesis. The number of patches are 4, 64, 25, and 4 respectively. The window size for metal artifact reduction is 80x 80 and the number of patches is 25.

As we can see, our method exhibits the lowest noise power across the spatial frequency, indicating the superiority of our proposed algorithm, which is in fact in line with the quantitative evaluation as reported in Table 1, 3, 5, and 7.

Figure 3. The noise power spectrum analysis (NPS) for different reconstruction tasks, including low-dose PET/CT reconstruction, metal artifact reduction, and accelerated MRI reconstruction by our framework. The error map between the reconstructed image and the GT image is regarded as the noise image for NPS calculation. The first row and second row correspond to two representative cases in Fig. 2.

Figure 5. The noise power spectrum analysis of the synthesized images by different state-of-the-art methods for the first two cases shown in Fig. 4. The error map between synthesized image and GT image is used as the noise map for NPS calculation.

Ref [1] Dobbins III, J. T., Samei, E., Ranger, N. T. & Chen, Y. Intercomparison of methods for image quality characterization. ii noise power spectrum. Med. physics 33, 1466–1475 (2006).

[Comment 4]:

The comparison methods with respect to reconstruction and MAR are too old. The authors should add advanced reconstruction methods to highlight the performance of proposed algorithm.

[Author’s answer and modification]:

Thank you for your comment. In the revised version, we have added the more recent MAR methods DuDoNet into comparison as the reviewer suggested. We summarize the quantitative performance in Table 3. As we can see that our proposed network based on ResUNet-32 backbone outperforms the other methods significantly in terms of both PSNR and SSIM.

Table 3. Quantitative comparison with representative methods for MAR on both in-house teeth CBCT dataset and public DeepLesion dataset in terms of SSIM \uparrow and PSNR ([dB]) \uparrow .

Method	CBCT		DeepLesion	
	SSIM	PSNR	SSIM	PSNR
LI ⁴⁵	0.8427 \pm 0.0954	31.75 \pm 2.89	0.9089 \pm 0.0983	30.72 \pm 2.75
NMAR ⁴⁶	0.9378 \pm 0.0812	31.54 \pm 2.68	0.9102 \pm 0.0889	31.89 \pm 2.01
CycleGAN ³⁶	0.8469 \pm 0.2234	31.55 \pm 2.97	0.7201 \pm 0.1403	30.72 \pm 1.97
RCN	0.9286 \pm 0.1865	33.60 \pm 1.69	0.9302 \pm 0.1351	33.81 \pm 1.44
AttentionMAR ⁴⁷	0.9507 \pm 0.0537	36.25 \pm 1.23	0.9421 \pm 0.1056	34.71 \pm 1.75
DuDoNet ²⁶	0.9572 \pm 0.0492	36.74 \pm 1.54	0.9492 \pm 0.1123	35.23 \pm 1.69
Ours (ResUNet-8)	0.9328 \pm 0.0573	35.02 \pm 1.36	0.9203 \pm 0.1518	33.76 \pm 2.82
Ours (ResUNet-16)	0.9599 \pm 0.0487	36.84 \pm 1.13	0.9327 \pm 0.1275	34.79 \pm 2.07
Ours (ResUNet-32)	0.9634 \pm 0.0295	36.91 \pm 1.084	0.9502 \pm 0.0872	35.93 \pm 1.99

[Comment 5]:

In Table 1, the metrics of PSNR and RMSE are the same meaning, the authors can give one of them.

[Author's answer and modification]:

Thanks for your comment. As your suggestion, we have removed the RMSE metric in the revised version to avoid redundancy.

[Comment 6]:

In respect to the table 2, the authors try to highlight the dual domain advantages over single domain, in fact, there are some great works have investigated. Therefore, this reviewer still concerns the originality of this paper.

[Author's answer and modification]:

We apologize for the unclarity of our previous draft. In the revised version, we have clearly clarified the difference between our proposed framework and the existing methods in the Discussion Section as highlighted in blue. In fact, our framework is built on dual domains, which originates from the medical imaging mechanism of the representative imaging systems such as MRI, CT, and PET. Unlike the others which usually adopt sequentially cascaded or parallel connected sub-networks for processing the individual domain patterns, we explicitly impose hierarchical consistency including intra-domain consistency, inter-domain consistency, and cycle consistency which are performed in three stages during the training phase. The stepwise consistency-constraint is able to achieve a stabilized and structured similarity match and hence an improved network performance.

Reply to Reviewer 2

Paper No: COMMS-23-0013

By: Jiadong Zhang, Kaicong Sun, Junwei Yang, Yan Hu, Yuning Gu, Zhiming Cui, Xiaopeng Zong, Fei Gao, Dinggang Shen.

Thank you very much for your affirmation and constructive comments on our paper. Your suggestions have helped us a lot to improve the quality of our manuscript. Based on your comments and suggestions, we have made the following revisions.

[Comment 1]:

In recent two years, many dual-domain deep learning based networks were reported for medical image reconstruction and synthesis. The authors may need a deep survey on this topic in the introduction to enhance the background of this paper.

[Author's answer and modification]:

We appreciate your constructive comment. In the modified manuscript, we have added additional literature survey including medical image reconstruction and synthesis in the Introduction Section as highlighted in blue. We copy the added contents as below:

“Different from the conventional CycleGAN framework, where cycle consistency is performed between the source and target images using unsupervised learning scheme in a single modality, e.g., the image domain, our proposed dual-domain based generative framework adopts the principle of hierarchical consistency in dual domains based on supervised learning.”

“More importantly, unlike most of the existing dual-domain based generative methods which either adopt sequentially cascaded or parallel connected sub-networks for processing the individual domain patterns, we explicitly impose hierarchical consistency including intra-domain consistency, inter-domain consistency, and cycle consistency.”

[Comment 2]:

For all the experimental results, in Tables 1-8, both PSNR and NRMSE are used. However, PSNR are identical to NRMSE up to a log operation and a constant bias. The authors may consider keep one of them to remove the redundancy.

[Author's answer and modification]:

Thanks for your comment. In our revised version, as the reviewer suggested, we have removed the RMSE metric to avoid redundancy.

[Comment 3]:

Regarding the ablation study, in Tables 2, 4, 6 and 8, to demonstrate that all the components are necessary, experiments should be performed for all the settings without the testing component. Hence, the authors may consider the settings “S2+S3” and “S1+S3”.

[Author’s answer and modification]:

Thank you for your comments. In the revised manuscript, we have added the additional experiment settings as the reviewer suggested for ablation study in all the investigated applications including low-dose PET/CT reconstruction, metal artifact reduction, MRI reconstruction, and PET-to-CT/CT-to-PET synthesis in Table 2, 4, 6, 8, respectively. According to the more comprehensive analysis, we better demonstrate the effectiveness of our proposed training strategy.

Table 2. Ablation study of the proposed framework for low-dose PET/CT reconstruction including multi-stage and dual-domain schemes in terms of SSIM \uparrow , PSNR ([dB]) \uparrow , SUV_{mean} (for PET images), and SUV_{max} (for PET images). S1, S2, and S3 represent three stages containing the hierarchical consistency constraints including intra-domain consistency, inter-domain consistency, and cycle consistency, respectively.

Setting	Low-dose PET reconstruction		Low-dose CT reconstruction	
	SSIM	PSNR	SSIM	PSNR
S1	0.8064 \pm 0.1028	35.07 \pm 1.03	0.9311 \pm 0.0823	41.02 \pm 2.31
S1+S2	0.9788 \pm 0.0274	37.22 \pm 1.70	0.9771 \pm 0.0138	42.02 \pm 0.91
S1+S3	0.9816 \pm 0.0225	37.51 \pm 1.49	0.9762 \pm 0.0167	42.49 \pm 0.95
S2+S3	0.9837 \pm 0.0201	38.25 \pm 1.96	0.9835 \pm 0.0137	42.95 \pm 0.98
S1+S2+S3	0.9939 \pm 0.0124	39.57 \pm 1.03	0.9910 \pm 0.0032	43.57 \pm 0.85
Image domain	0.9815 \pm 0.0139	37.02 \pm 1.41	0.9666 \pm 0.0172	41.94 \pm 1.81
Dual domain	0.9939 \pm 0.0124	39.57 \pm 1.03	0.9910 \pm 0.0032	43.57 \pm 0.85

Table 4. Ablation study of the proposed framework for MAR on both in-house teeth CBCT dataset and public DeepLesion dataset in terms of SSIM \uparrow and PSNR ([dB]) \uparrow .

Setting	CBCT		DeepLesion	
	SSIM	PSNR	SSIM	PSNR
S1	0.9303 \pm 0.0812	34.28 \pm 1.954	0.9305 \pm 0.2433	32.85 \pm 2.78
S1+S2	0.9527 \pm 0.0534	35.86 \pm 1.379	0.9396 \pm 0.2153	34.71 \pm 1.88
S1+S3	0.9579 \pm 0.0572	36.35 \pm 1.582	0.9427 \pm 0.2272	35.02 \pm 2.15
S2+S3	0.9531 \pm 0.0423	36.21 \pm 1.864	0.9404 \pm 0.1955	34.88 \pm 2.32
S1+S2+S3	0.9634 \pm 0.0295	36.91 \pm 1.084	0.9502 \pm 0.0872	35.93 \pm 1.99
Image domain	0.9601 \pm 0.0563	35.87 \pm 1.563	0.9371 \pm 0.1732	34.35 \pm 2.30
Dual domain	0.9634 \pm 0.0295	36.91 \pm 1.084	0.9502 \pm 0.0872	35.93 \pm 1.99

Table 6. Ablation study of the proposed framework for MRI reconstruction based on the UNet backbone on in-house dataset for 4 \times and 8 \times acceleration rates in terms of SSIM \uparrow and PSNR ([dB]) \uparrow .

Setting	4 \times		8 \times	
	SSIM	PSNR	SSIM	PSNR
S1	0.9808 \pm 0.0061	37.05 \pm 1.89	0.9734 \pm 0.0093	35.31 \pm 2.23
S1+S2	0.9862 \pm 0.0042	39.13 \pm 1.81	0.9762 \pm 0.0085	35.75 \pm 2.49
S1+S3	0.9862 \pm 0.0036	39.48 \pm 1.64	0.9781 \pm 0.0069	36.04 \pm 1.71
S2+S3	0.9863 \pm 0.0038	39.50 \pm 1.71	0.9781 \pm 0.0062	35.87 \pm 1.38
S1+S2+S3	0.9866 \pm 0.0041	39.53 \pm 1.73	0.9782 \pm 0.0069	36.88 \pm 1.80
Image domain	0.9808 \pm 0.0061	37.05 \pm 1.89	0.9734 \pm 0.0093	35.31 \pm 2.23
Dual domain	0.9866 \pm 0.0041	39.53 \pm 1.73	0.9782 \pm 0.0069	36.88 \pm 1.80

Table 8. Ablation study of the proposed framework for PET-CT synthesis on in-house dataset in terms of SSIM \uparrow , PSNR (dB) \uparrow , SUV_{mean} (for PET images), and SUV_{max} (for PET images).

Setting	PET-to-CT		CT-to-PET	
	SSIM	PSNR	SSIM	PSNR
S1	0.9569 \pm 0.0484	35.54 \pm 1.02	0.9491 \pm 0.0390	34.82 \pm 1.62
S1+S2	0.9810 \pm 0.0319	37.30 \pm 1.21	0.9612 \pm 0.0312	36.02 \pm 1.27
S1+S3	0.9573 \pm 0.0482	37.15 \pm 1.76	0.9486 \pm 0.0582	35.66 \pm 1.89
S2+S3	0.9828 \pm 0.0327	38.01 \pm 1.11	0.9384 \pm 0.0598	36.21 \pm 1.58
S1+S2+S3	0.9843 \pm 0.0123	38.33 \pm 0.55	0.9658 \pm 0.0436	36.76 \pm 1.25
Image domain	0.9783 \pm 0.0155	36.24 \pm 1.01	0.9379 \pm 0.0759	34.84 \pm 1.96
Dual domain	0.9843 \pm 0.0123	38.33 \pm 0.55	0.9658 \pm 0.0436	36.76 \pm 1.25

[Comment 4]:

In all the tables, the values of SSIM are close to 1.0, and almost all the values are greater than 0.95. The differences between the proposed method and the competing methods are very small. Noting that the default parameters in the SSIM function were designed for nature images with a pixel value range of 0-255. Because the original

[Author’s answer and modification]:

Thank you for your comments regarding our evaluation of image quality. In our study, we utilize the parameter settings for the SSIM metric as specified in the paper by Wang *et al.* [1], which is in fact widely used in many representative works of medical image reconstruction or synthesis tasks [2,3,4]. Since the established parameter setting (K1 = 0.01, K2 = 0.03) has been widely adopted in the medical image processing community, by adhering to the parameters defined in [1], we aim to maintain consistency and ensure comparability with the existing literature. Although we acknowledge that the current parameter setting in SSIM may not describe the performance difference of different methods in the best way, these recommended setting is too widely used and considered as “standard” in our field. Using customized parameter setting of SSIM could introduce bias in our evaluation and may also make it difficult for the others to directly compare with our results on public datasets such as DeepLesion dataset for MAR.

[1] Wang, Zhou, et al. "Image quality assessment: from error visibility to structural similarity." *IEEE transactions on image processing* 13.4 (2004): 600-612.

[2] Hyun, Chang Min, et al. "Deep learning for undersampled MRI reconstruction." *Physics in Medicine & Biology* 63.13 (2018): 135007.

[3] Wang, Yan, et al. "3D auto-context-based locality adaptive multi-modality GANs for PET synthesis." *IEEE transactions on medical imaging* 38.6 (2018): 1328-1339.

[4] Zhang, Yanbo, and Hengyong Yu. "Convolutional neural network based metal artifact reduction in x-ray computed tomography." *IEEE transactions on medical imaging* 37.6 (2018): 1370-1381.

[Comment 5]:

For a deep learning based method, network training is time-consuming. For all the deep learning based methods in each experiment, it is better to clarify the computational cost to train the proposed network framework and the related competing networks.

[Author’s answer and modification]:

Thank you for your comments. In the revised manuscript, we have detailed the required training time of all the investigated methods for different applications as marked in blue in the Implementation Details Section.

[Comment 6]:

Regarding the application of low-dose PET/CT reconstruction, all the competing methods listed in Table 1 only use image-domain information. Since the proposed method is in dual-domain, at least one state-of-the-art dual-domain image reconstruction approach should be compared. For example: Wu et al, DRONE: Dual-domain residual-based optimization network for sparse-view CT reconstruction, IEEE-TMI, 40(11):3002-3014, 2021.

Jiao et al. A dual-domain CNN-based network for CT reconstruction. IEEE Access.

[Author’s answer and modification]:

Thanks for your comment. In the revised manuscript, we have added additional experiment to compare our method with the recent dual-domain reconstruction framework iBP-Net [1] as the reviewer suggested. We summarize the quantitative evaluation in Table 1. We can see that although iBP-Net exploits both domains and obtains better reconstruction performance than other comparative methods, it cannot explicitly guarantee dual-domain consistency. In contrast, our method uses elaborately designed consistency-constraints and three-stage training scheme, and hence outperforms the investigated methods significantly.

Table 1. Quantitative comparison with representative learning-based methods for low-dose PET/CT reconstruction in terms of SSIM \uparrow , PSNR (dB) \uparrow , and NRMSE ($\times 10^{-2}$) \downarrow .

Tasks Method	Low-dose PET reconstruction			Low-dose CT reconstruction		
	SSIM	PSNR	NRMSE	SSIM	PSNR	NRMSE
RU-Net ³²	0.8064 \pm 0.1028	35.07 \pm 1.03	20.99 \pm 8.19	0.9311 \pm 0.0823	41.02 \pm 2.31	7.74 \pm 6.40
p2pGAN ³³	0.9844 \pm 0.0129	37.55 \pm 1.20	12.12 \pm 6.08	0.9712 \pm 0.0510	42.41 \pm 1.79	5.17 \pm 3.60
CycleGAN ²⁹	0.9861 \pm 0.0078	37.85 \pm 1.18	10.90 \pm 3.17	0.9811 \pm 0.0086	41.90 \pm 0.94	5.10 \pm 0.74
MedGAN ²⁷	0.9863 \pm 0.0093	37.72 \pm 1.11	11.52 \pm 5.41	0.9613 \pm 0.0577	41.53 \pm 2.05	6.51 \pm 4.73
iBP-Net ²⁷	0.9859 \pm 0.0154	38.47 \pm 1.28	10.31 \pm 4.58	0.9812 \pm 0.0143	42.71 \pm 0.97	4.26 \pm 0.85
Ours	0.9939 \pm 0.0124	39.57 \pm 1.03	6.35 \pm 1.48	0.9910 \pm 0.0032	43.57 \pm 0.85	3.46 \pm 0.43

[1] Jiao et al, A dual-domain CNN-based network for CT reconstruction, IEEE Access, 9:71091-71103, 2021.

[Comment 7]:

Regarding the application of metal artifact reduction, at least one state-of-the-art dual-domain approach should be compared. For example, references 18 and 22 cited in this paper.

[Author’s answer and modification]:

Thanks for your suggestion. In the revised manuscript, we have added additional experiments using the more recent method DoDuNet as the reviewer suggested. We summarize the quantitative evaluation on both private and public datasets in Table 3. It is shown that our method outperforms the DoDuNet also in terms of both PSNR and SSIM.

Table 3. Quantitative comparison with representative methods for MAR on both in-house teeth CBCT dataset and public DeepLesion dataset in terms of SSIM \uparrow and PSNR (dB) \uparrow .

Method	CBCT		DeepLesion	
	SSIM	PSNR	SSIM	PSNR
LI ⁴⁵	0.8427 \pm 0.0954	31.75 \pm 2.89	0.9089 \pm 0.0983	30.72 \pm 2.75
NMAR ⁴⁶	0.9378 \pm 0.0812	31.54 \pm 2.68	0.9102 \pm 0.0889	31.89 \pm 2.01
CycleGAN ³⁶	0.8469 \pm 0.2234	31.55 \pm 2.97	0.7201 \pm 0.1403	30.72 \pm 1.97
RCN	0.9286 \pm 0.1865	33.60 \pm 1.69	0.9302 \pm 0.1351	33.81 \pm 1.44
AttentionMAR ⁴⁷	0.9507 \pm 0.0537	36.25 \pm 1.23	0.9421 \pm 0.1056	34.71 \pm 1.75
DuDoNet ²⁶	0.9572 \pm 0.0492	36.74 \pm 1.54	0.9492 \pm 0.1123	35.23 \pm 1.69
Ours (ResUNet-8)	0.9328 \pm 0.0573	35.02 \pm 1.36	0.9203 \pm 0.1518	33.76 \pm 2.82
Ours (ResUNet-16)	0.9599 \pm 0.0487	36.84 \pm 1.13	0.9327 \pm 0.1275	34.79 \pm 2.07
Ours (ResUNet-32)	0.9634 \pm 0.0295	36.91 \pm 1.084	0.9502 \pm 0.0872	35.93 \pm 1.99

[Comment 8]:

Regarding the application of fast MRI reconstruction, at least one state-of-the-art dual-domain approach should be compared. For example, references 19 and 21 cited in this paper.

[Author’s answer and modification]:

Thanks for your suggestion. In the revised manuscript, we have added additional experiments using reference 19 (DuDoRNet) according to the reviewer’s comment. Experimental results are summarized in Table 5. We can see that our method outperforms DuDoRNet on both datasets and both subsampling rates in terms of SSIM and PSNR.

Table 5. Quantitative evaluation of our framework based on two representative backbones UNet and E2EVarNet using the in-house dataset for 4 \times and 8 \times acceleration rates in terms of SSIM \uparrow and PSNR (dB) \uparrow .

Network	4 \times		8 \times	
	SSIM	PSNR	SSIM	PSNR
UNet ²⁰	0.9808 \pm 0.0061	37.05 \pm 1.89	0.9734 \pm 0.0093	35.31 \pm 2.23
UNet (Ours)	0.9866 \pm 0.0041	39.53 \pm 1.73	0.9782 \pm 0.0069	36.88 \pm 1.80
E2EVarNet ²¹	0.9653 \pm 0.0073	36.12 \pm 1.64	0.9481 \pm 0.0121	33.29 \pm 1.41
E2EVarNet (Ours)	0.9703 \pm 0.0069	37.00 \pm 1.46	0.9561 \pm 0.0116	34.13 \pm 1.38
DuDoRNet ²⁰	0.9861 \pm 0.0055	38.08 \pm 2.21	0.9733 \pm 0.011	34.70 \pm 2.10

Reply to Reviewer 3

Paper No: COMMS-23-0013

By: Jiadong Zhang, Kaicong Sun, Junwei Yang, Yan Hu, Yuning Gu, Zhiming Cui, Xiaopeng Zong, Fei Gao, Dinggang Shen.

Dear Reviewer,

Thank you very much for your affirmation and constructive comments of our paper. Your suggestions have helped us a lot to improve the quality of our manuscript. Based on your comments and suggestions, we have made the following revisions.

[Comment 1]:

Just curious, in PET/CT reconstruction and PET-CT synthesis a fully connected network is employed for G_s^A and G_t^A , but in MAR, RU-Net is employed, and in fast MRI, E2EVarNet is employed. Are there any suggestions for the deployment of different architectures for different tasks? As a generalized framework, the method

[Author's answer and modification]:

Thank you for your constructive comments. In the revised manuscript, we have added some recommendations on the network backbone for different tasks as described in the Discussion Section marked in blue based on our experimental experience. In fact, one can always use UNet-shaped network as the baseline backbone for the individual generative functions G , and individually designed network structure for different applications can further facilitate the model performance.

In the MRI reconstruction task, as the reviewer suggested, we have used two backbones, i.e., UNet and VarNet, and we demonstrate the results in Table 6. Based on the experimental results, it is shown that using UNet as backbone performs better than the VarNet one.

[Comment 2]:

Seems that the same metric is used in all the experimental results. A more thorough or clinically related metric may be considered for further evaluation on one of the dataset, for example (SUVs) for PET [Ref1, Ref2] or radiologist report for the synthesized images.

[Author's answer and modification]:

Thanks for your comment. In the revised version, we have calculated the SUV mean bias and SUV max bias as the reviewer suggested for synthesized PET evaluation in both low-dose PET reconstruction and CT-to-PET synthesis tasks. The SUV bias is calculated via following equation:

$$Bias = \frac{SUV_{mean/max}^S - SUV_{mean/max}^R}{SUV_{mean/max}^R} \times 100\%$$

where $SUV_{mean/max}^R$ is the mean or max SUV of ground-truth PET images and $SUV_{mean/max}^S$ is the mean or max SUV of synthesized PET images. We demonstrate the performance of different methods in Table 1 and Table 7, and we can see that our method has the least SUV bias compared with other methods, indicating great advantages of our method.

Table 1. Quantitative comparison with representative learning-based methods for low-dose PET/CT reconstruction in terms of SSIM \uparrow , PSNR ([dB]) \uparrow , SUV_{mean} (for PET images), and SUV_{max} (for PET images).

Tasks Method	Low-dose PET reconstruction				Low-dose CT reconstruction	
	SSIM	PSNR	SUV_{mean}	SUV_{max}	SSIM	PSNR
RUNet ⁴⁰	0.8064 \pm 0.1028	35.07 \pm 1.03	8.70 \pm 6.79	6.23 \pm 5.94	0.9311 \pm 0.0823	41.02 \pm 2.31
p2pGAN ⁴¹	0.9844 \pm 0.0129	37.55 \pm 1.20	5.99 \pm 9.04	3.92 \pm 5.02	0.9712 \pm 0.0510	42.41 \pm 1.79
CycleGAN ³⁷	0.9861 \pm 0.0078	37.85 \pm 1.18	7.79 \pm 5.86	-6.63 \pm 1.04	0.9811 \pm 0.0086	41.90 \pm 0.94
MedGAN ³⁵	0.9863 \pm 0.0093	37.72 \pm 1.11	4.01 \pm 8.16	-8.31 \pm 4.59	0.9613 \pm 0.0577	41.53 \pm 2.05
iBP-Net ⁴²	0.9859 \pm 0.0154	38.47 \pm 1.28	6.84 \pm 3.97	5.96 \pm 1.79	0.9812 \pm 0.0143	42.71 \pm 0.97
Ours	0.9939 \pm 0.0124	39.57 \pm 1.03	3.62 \pm 5.83	3.54 \pm 2.40	0.9910 \pm 0.0032	43.57 \pm 0.85

Table 7. Quantitative comparison with other state-of-the-art methods for PET-CT synthesis based on in-house dataset in terms of SSIM \uparrow , PSNR ([dB]) \uparrow , SUV_{mean} (for PET images), and SUV_{max} (for PET images).

Tasks Method	PET-to-CT			CT-to-PET		
	SSIM	SSIM	PSNR	PSNR	SUV_{mean}	SUV_{max}
U-Net ⁴³	0.9581 \pm 0.0427	35.10 \pm 1.06	0.9160 \pm 0.0357	33.52 \pm 1.05	9.02 \pm 4.22	-7.48 \pm 6.86
RUNet ⁴⁰	0.9569 \pm 0.0484	35.54 \pm 1.02	0.9491 \pm 0.0390	34.82 \pm 1.62	8.12 \pm 2.74	-9.62 \pm 1.07
p2pGAN ⁴¹	0.9666 \pm 0.0200	35.18 \pm 0.98	0.9391 \pm 0.0484	35.04 \pm 1.67	9.52 \pm 3.19	-9.36 \pm 1.90
CycleGAN ³⁷	0.9646 \pm 0.0197	35.49 \pm 1.04	0.9581 \pm 0.0402	35.70 \pm 2.01	8.86 \pm 6.43	-8.06 \pm 2.81
MedGAN ³⁵	0.9706 \pm 0.0111	35.70 \pm 0.64	0.9091 \pm 0.0658	35.83 \pm 2.77	7.36 \pm 7.58	-5.04 \pm 2.07
Ours	0.9843 \pm 0.0123	38.33 \pm 0.55	0.9658 \pm 0.0436	36.76 \pm 1.25	4.98 \pm 3.07	3.27 \pm 4.46

REVIEWERS' COMMENTS:

Reviewer #1 (Remarks to the Author):

Satisfied with the response.

Reviewer #2 (Remarks to the Author):

All of my concerns have been addressed. Hence, I would like to recommend it for publication.

Reviewer #3 (Remarks to the Author):

The authors have addressed all my concerns.

The revised manuscript is now of high quality to me and presents a solid contribution to the field. I believe that the paper is in excellent shape for publication in this esteemed journal.